# The Relationship between Elevated Homocysteine and Metabolic Syndrome in a Community-Dwelling Middle-Aged and Elderly Population in Taiwan

**DOI:** 10.3390/biomedicines11020378

**Published:** 2023-01-27

**Authors:** Yu-Lin Shih, Chin-Chuan Shih, Tzu-Cheng Huang, Jau-Yuan Chen

**Affiliations:** 1Department of Family Medicine, Chang-Gung Memorial Hospital, Linkou Branch, Taoyuan 333, Taiwan; 2United Safety Medical Group, General Administrative Department, New Taipei City 242, Taiwan; 3College of Medicine, Chang Gung University, Taoyuan 333, Taiwan

**Keywords:** homocysteine level, metabolic syndrome, middle-aged and elderly, obesity, waist circumference

## Abstract

(1) Background: Metabolic syndrome has become a serious health problem in society. Homocysteine is a biomarker for cardiovascular disease. We investigated the relationship between homocysteine levels and metabolic syndrome. (2) Methods: A total of 398 middle-aged and elderly individuals were included in our study. First, we divided the participants into two groups: the metabolic syndrome group and the nonmetabolic syndrome group. Second, according to tertiles of homocysteine levels from low to high, the participants were divided into first, second, and third groups. Pearson’s correlation was then calculated for homocysteine levels and metabolic factors. Scatterplots are presented. Finally, the risk of metabolic syndrome in the second and third groups compared with the first group was assessed by multivariate logistic regression. (3) Results: In our study, the metabolic syndrome group had higher homocysteine levels, and the participants in the third group were more likely to have metabolic syndrome. Multivariate logistic regression revealed that the third group, which had the highest homocysteine level, was associated with metabolic syndrome with an odds ratio of 2.32 compared with the first group after adjusting for risk factors. (4) Conclusions: We concluded that high plasma homocysteine levels were independently associated with MetS in our study population.

## 1. Introduction

Metabolic syndrome has become a global health problem and consumes tremendous amounts of medical resources. Metabolic syndrome is a cluster of metabolic abnormalities, including obesity, glucose intolerance, elevated blood pressure (BP), and dyslipidemia [1], leading to a higher risk of metabolic diseases and cardiovascular diseases [2,3]. Many theories explain the association between metabolic syndrome and cardiovascular diseases. Metabolic syndrome criteria include dyslipidemia, which can cause atherosclerosis and lead to many cardiovascular diseases. The role of lipids, including cholesterol, in cardiovascular diseases is well-developed [4]. However, fat tissue can be seen as potent endocrine tissue since adipocytes can produce many bioactive agents to regulate the homeostasis of the body. Inflammatory cytokines produced by adipocytes increase oxidative stress in the endothelium and compromise circulation [5]. Homocysteine is a novel biomarker of cardiovascular disease and may play an important role in the association between metabolic syndrome and cardiovascular diseases. It is believed that elevated homocysteine levels damage endothelial cells [6]. A recent study revealed that elevated insulin can disturb the metabolism of homocysteine and cause elevated homocysteine levels [7]. Insulin resistance in metabolic syndrome is marked by high insulin levels [8]. This evidence indicates that homocysteine may play an important role in the relationship between metabolic syndrome and cardiovascular disease. Additionally, elderly populations are vulnerable to cardiometabolic diseases [9]. Hence, in our study, we investigated the relationship between homocysteine levels and metabolic syndrome among the middle-aged and elderly population in Taiwan.

## 2. Materials and Methods

### 2.1. Study Design and Participants

This was a cross-sectional community-based study based on a community health promotion project conducted in northern Taiwan in 2019. Figure 1 shows the flowchart of recruitment. Initially, 1065 participants were recruited. The inclusion criteria of participants were as follows: (1) aged 50−85 years; (2) lived in the community and able to walk to the clinic; (3) able to complete a questionnaire; and (4) completed all surveys. Some participants were excluded based on the following criteria: (1) a history of recent heart disease; (2) unable to complete all examinations; and (3) declined to participate. We recruited 396 eligible participants for analysis. This study was approved by the Chang Gung Medical Foundation Institutional Review Board (IRB, No. 201801803B0), and consent was obtained from all participants prior to enrollment.

### 2.2. Data Collection

The content of the questionnaire included age, sex, alcohol consumption habits (current drinker or not), and current smoking habits (current smoker or not). Data on alcohol consumption and smoking habits were both self-reported. Health survey data were collected in the study, including blood pressure (BP, mmHg), body mass index (BMI), and biochemical laboratory data. Systolic blood pressure (SBP, mmHg) and diastolic blood pressure (DBP, mmHg) were measured at least twice at rest. Waist circumference was measured by measuring the midpoint of the iliac crest and the last rib in a horizontal plane while the participants were in a standing position. The biochemical laboratory data were analyzed at the Roche model lab of the Taiwan E&Q Clinical Laboratory. The collected biochemical laboratory data included levels of homocysteine (µmol/L), fasting plasma glucose (FPG, mg/dL), alanine transaminase (ALT, mg/dL), creatinine (mg/dL), high-density lipoprotein cholesterol (HDL-C, mg/dL), low-density lipoprotein cholesterol (LDL-C, mg/dL), triglycerides (mg/dL), and uric acid (UA, mg/dL). We defined hypertension as undergoing treatment for hypertension, SBP ≥ 140 mmHg, DBP ≥ 90 mmHg, or a history of hypertension [10]. We defined diabetes mellitus as undergoing insulin therapy, oral hypoglycemic agents, FPG ≥ 126 mg/dL, or a history of diabetes mellitus [11]. Dyslipidemia was defined as the use of lipid-lowering medication, HDL-C < 40 mg/dL in men or <50 mg/dL in women, LDL-C ≥ 130 mg/dL, total cholesterol ≥ 200 mg/dL, triglycerides ≥ 150 mg/dL, or a history of dyslipidemia [12].

### 2.3. Definition of Metabolic Syndrome and Homocysteine Levels

In this study, metabolic syndrome was defined [13] by the presence of three or more of the following components: (1) WC ≥ 90 cm for men and ≥80 cm for women; (2) triglycerides ≥ 150 mg/dL; (3) HDL-C < 40 mg/dL for men and <50 mg/dL for women; (4) SBP/DBP ≥ 130/85 mmHg or current use of antihypertensive medications; and (5) FPG ≥ 100 mg/dL. Homocysteine levels were measured at the Taiwan E&Q Clinical Laboratory. The participants were categorized into the first group (homocysteine < 11.1 µmol/L), second group (homocysteine between 11.1 and 14.3 µmol/L), and third group (homocysteine ≥ 14.4 µmol/L) according to the tertile of homocysteine levels from low too high in our study.

### 2.4. Statistical Analysis

We divided the participants into two groups according to metabolic syndrome status. Laboratory and anthropometric data were analyzed. Categorical variables were analyzed using the chi-square test and are expressed as n (%). Continuous variables were analyzed using Student’s *t*-test and are expressed as the means ± SD. Then, we divided all enrolled participants into three groups: the first group (homocysteine < 11.1 µmol/L), the second group (homocysteine between 11.1 and 14.3 µmol/L), and the third group (homocysteine ≥ 14.4 µmol/L). Categorical variables were analyzed using the chi-square test and expressed as *n* (%). Continuous variables were analyzed using analysis of variance (ANOVA) and expressed as the means ± SD. Pearson’s correlation test was performed to analyze the correlations between homocysteine and FPG, waist circumference, SBP, DBP, triglycerides, and HDL-C. A scatterplot of homocysteine levels by waist circumference, SBP, DBP, and HDL-C was used to demonstrate the relationship. To analyze the association between homocysteine levels and metabolic syndrome, multiple logistic regression was performed. Three regression models were used. Model 1 was unadjusted. Age and sex were adjusted in Model 2. Age, sex, smoking, drinking, triglycerides, systolic blood pressure, and fasting plasma glucose levels were adjusted in Model 3. In our study, we defined a *p*-value of <0.05 as statistically significant. SPSS for Windows (IBM Corp. Released 2011; IBM SPSS Statistics, version 20.0. Armonk, NY, USA: IBM Corp.) was used for statistical analyses.

## 3. Results

### 3.1. The Characteristics of Study Subjects by Metabolic Syndrome Status

This study included middle-aged and elderly people recruited from local medical clinics in northern Taiwan. The enrolled patients consisted of 164 men (41.4%) and 232 women (58.6%) with a mean age of 64.80 ± 8.77 years. In total, 396 participants were included. Clinical and anthropometric data are summarized in Table 1. The enrolled patients were divided into two groups, the metabolic syndrome group and the nonmetabolic syndrome group, based on whether they met the criteria for metabolic syndrome. To compare the clinical data between the metabolic syndrome group and the nonmetabolic syndrome group, Student’s *t*-test and the chi-square test were performed. The discrepancy in each clinical data point between the two groups is shown in Table 1. Higher homocysteine levels were found in the metabolic syndrome group with statistical significance. In addition, the participants with metabolic syndrome tended to have higher levels of FPG, ALT, creatinine, triglycerides, UA, AC, and BMI. The enrolled patients in the metabolic syndrome group tended to have drinking habits, hypertension, diabetes mellitus, and dyslipidemia with statistical significance. In addition, lower HDL-C levels and LDL-C levels were found in the metabolic syndrome group. No statistically significant differences were found between the metabolic syndrome group and nonmetabolic syndrome group regarding systolic blood pressure (SBP), diastolic blood pressure (DBP), age, smoking rate, and gender. Based on the definition of metabolic syndrome [7], we compared the criteria of metabolic syndrome in both groups using the chi-square test. The results are shown in Table 2. As we expected, the metabolic syndrome group had significantly higher WC, BP, FPG, and triglycerides than the nonmetabolic group. In addition, the metabolic group had significantly lower HDL-C levels than the nonmetabolic syndrome group.

### 3.2. The Characteristics of Study Subjects According to Tertiles of Homocysteine

Table 3 shows the anthropometric data and indices related to metabolic syndrome. The mean homocysteine level of the study group was 13.60 ± 4.90. The participants were separated into three groups according to the three tertiles of homocysteine levels. No statistically significant differences were found among the three groups in ALT levels, LDL-C levels, triglyceride levels, age, BMI, alcohol consumption, and dyslipidemia rate. However, the participants in the high homocysteine group tended to have significantly higher levels of FPG, creatinine, HDL-C, UA, WC, SBP, and DBP. In addition, the enrolled subjects in the high homocysteine group tended to have a higher prevalence of smoking, hypertension, diabetes mellitus, and metabolic syndrome and were more likely to be men. High-density lipoprotein cholesterol levels were lower in the high homocysteine group.

### 3.3. Pearson’s Correlation between Homocysteine and Components of Metabolic Syndrome

Table 4 summarizes the correlations between homocysteine levels and components of metabolic syndrome. Positive correlations were also observed between homocysteine levels and WC, SBP, and DBP, with statistical significance. A statistically significant inverse correlation was found between homocysteine levels and HDL-C levels. There was no statistically significant correlation between homocysteine levels and FPG and triglyceride levels. Scatterplots with the results of Pearson’s correlation of homocysteine levels by waist circumference, SBP, DBP, and HDL-C are shown in Figure 2, Figure 3, Figure 4 and Figure 5, respectively.

### 3.4. Multiple Logistic Regression Analysis of the Association between Homocysteine Level and Metabolic Syndrome

In Table 5, to further investigate the association between homocysteine and metabolic syndrome, multivariate logistic regression models were used to calculate the odds ratio (OR) of homocysteine levels with metabolic syndrome after adjustment for other risk factors. The second and third groups were compared with the first group. Three models were applied. Model 1 was unadjusted; Model 2 was adjusted for age and sex; and Model 3 was adjusted for age, sex, smoking, drinking, triglycerides, SBP, and FPG. There was no significant difference between the middle and low homocysteine groups across all three models. In comparison, logistic regression showed that the OR of the third group was significantly higher than the OR of the first group in all three models. The odds ratio for metabolic syndrome was 2.32 (1.12 to 4.78), with a *p*-value of 0.02 in the third group compared to the first group in Model 3.

## 4. Discussion

From previous medical studies, metabolic syndrome has a strong relationship with many chronic diseases, including hypertension [14], diabetes mellitus [15], and dyslipidemia [16]. Chronic diseases related to metabolic syndrome have posed a long-term healthcare burden worldwide. In this community-based study, we investigated the positive relationship between homocysteine levels and metabolic syndrome in middle-aged and elderly people in Taiwan. Considering the difference between metabolic syndrome and nonmetabolic syndrome (Table 1), there was a high level of FPG, triglycerides, waist circumference, SBP, and DBP in the metabolic syndrome group. These results correspond with those of previous studies [17,18,19]. Additionally, the metabolic syndrome group had lower HDL levels, which was noted in a previous study [20]. This finding corresponds with the risk factors for metabolic syndrome. FPG, triglycerides, waist circumference, SBP, and DBP are risk factors for metabolic syndrome, but HDL serves as a protective factor against metabolic syndrome [20]. Moreover, hypertension, diabetes mellitus, and dyslipidemia were highly prevalent in the metabolic syndrome group. Those chronic diseases that showed a strong relationship with metabolic syndrome were also mentioned in previous studies [17,18,19]. In Table 2, we compare the criteria of metabolic syndrome [13] between the two groups. As we expected, the metabolic syndrome group tended to have higher WC, BP, FPG, and triglycerides than the nonmetabolic syndrome group. In addition, the metabolic group tended to have lower HDL-C levels. Overall, the results in Table 1 and Table 2 demonstrate a common profile of metabolic syndrome. Additionally, in Table 1, we noted kidney and liver dysfunction in the metabolic syndrome group. ALT, creatinine, and UA levels were elevated in the metabolic syndrome group, which was reported in previous studies [21,22,23]. Importantly, we noted a statistically significant increase in homocysteine levels in the metabolic syndrome group. This result evoked our curiosity about the relationship between homocysteine levels and metabolic syndrome.

In Table 3, we further discuss the factors of metabolic syndrome and other parameters in the three homocysteine tertiles. There was an increase in FPG, WC, SBP, DBP, and triglycerides as homocysteine levels increased. These results were also reported in previous studies [24,25,26]; however, HDL had a reverse relationship with homocysteine levels, which was also noted in a previous study [27]. The proportion of chronic diseases, including hypertension and diabetes mellitus, which have a strong relationship with metabolic syndrome, also increased as homocysteine levels increased, and the results of previous studies support this finding [25,26]. According to the results of our study, which indicated a strong relationship between homocysteine levels and metabolic syndrome, it is not surprising that the proportion of metabolic syndrome was elevated as homocysteine levels increased.

Table 4 focuses on the relationship between parameters in the metabolic syndrome definition and homocysteine levels. Pearson’s correlation coefficients are shown in Table 4. Homocysteine levels were positively correlated with FPG, WC, SBP, DBP, and triglycerides. HDL has a negative relationship with homocysteine levels. Among these correlations, only waist circumference, SBP, DBP, and HDL-C were statistically significant. In Figure 2, Figure 3, Figure 4 and Figure 5, trends in the scatterplot of homocysteine levels by waist circumference, SBP, DBP, and HDL-C are observed.

There are many factors that can impact homocysteine and metabolic syndrome; therefore, logistic regression was used (Table 5). The second and third groups were compared with the first group, and the prevalence of metabolic syndrome increased as homocysteine levels increased. After adjusting for age, sex, smoking, alcohol consumption, triglycerides, SBP, and FPG, the OR for metabolic syndrome was 2.32, with a 95% confidence interval between 1.12 and 4.78 in the third group compared with that in the first group. This result confirmed that homocysteine levels are an independent risk factor for metabolic syndrome.

As many countries have become wealthy enough to obtain sufficient food, metabolic syndrome has become a health issue in these countries. According to the National Cholesterol Education Panel, Adult Treatment Panel III (NCEP ATP III) [28], a diagnosis of metabolic syndrome is established if the patient has three or more out of five of the following: an increase in waist circumference (WC > 102 cm for men and >88 cm for women), blood pressure (BP > 135/85 mm/Hg), fasting blood glucose (FPG > 6.1 mmol/L), and triglycerides (TG > 1.7 mmol/L), and a decrease in high-density lipoprotein cholesterol levels (HDL-C < 1.03 mmol/L for men and HDL-C < 1.29 mmol/L for women). Our findings correspond with the criteria of ATP III. Additionally, we found that hyperhomocysteinemia was associated with the criteria for metabolic syndrome. We would like to further discuss the relationship between metabolic syndrome and homocysteine.

Metabolic syndrome is a problem of epidemic proportions, increasing the risk for dyslipidemia, diabetes mellitus type 2, atherosclerosis, cardiovascular diseases, and other chronic diseases [14,16,17,18]. Excessive adipose tissue alters the physiological function of patients with metabolic syndrome because adipose tissue is a highly active endocrine tissue. Many factors, such as insulin resistance, leptin, plasminogen activation inhibitor, adiponectin, adipokines, and other inflammatory cytokines, can be regulated by adipose tissue [29,30]. Chronic inflammation and factors that are closely related to adipose tissue are involved in the development of atherosclerosis and cardiovascular disease (CVD) [25]. According to statistics, 20–30% of the global population has metabolic syndrome [26], which increases the risk of cardiovascular mortality and the risk of myocardial infarction or stroke [31,32]. All the evidence above indicates the mechanism underlying the relationship between metabolic syndrome and the cardiovascular system.

Homocysteine is considered a potentially harmful agent that can compromise macrocirculation and microcirculation [6]. Numerous studies have revealed the association between endothelial dysfunction and homocysteine, which increases oxidative stress and injures the vascular endothelium [33]. Homocysteine decreases the level of nitrogen oxide, increases the proliferation of vascular smooth cells, and alters the elastic property of the vascular wall [34]. The compromised vascular wall contributes to the development of hypertension [35]. Similarly, we found that hypertension was related to hyperhomocysteinemia in our study. Hyperhomocysteinemia also leads to atherosclerosis, aneurysm, the development of cardiac hypertrophy, and other cardiovascular diseases [36,37]. Moreover, elevated homocysteine may damage the microcirculation in the kidney [38], potentially leading to chronic kidney disease.

Homocysteine might play a key role in the association between metabolic syndrome and cardiovascular disease because insulin resistance can affect the metabolism of homocysteine [39]. The Biosynthesis of homocysteine is a multi-step process. Homocysteine is made from methionine. First, S-adenosyl-methionine synthetase transfers the adenosine group from adenosine triphosphate (ATP) to methionine to yield S-adenosyl methionine (SAM-e). Then, SAM-e transfers a methyl group to other molecules, and the adenosyl group is hydrolyzed to homocysteine [40]. Homocysteine either becomes cysteine or is recycled into methionine [41]. In mammals, cysteine is made from homocysteine. The first step is that homocysteine combines with serine to yield cystathionine, and this step is catalyzed by cystathionine β-synthase (CBS). In the second step, cystathionine γ-lyase (CSE) converts cystathionine to ammonia, cysteine, and α-ketobutyrate [42]. The brain, liver, kidney, and pancreas have abundant CBS [43]. In a previous study, insulin repressed CBS expression [44]. Insulin resistance is an important feature of metabolic syndrome and leads to hyperinsulinemia in most patients with metabolic syndrome [45]. Elevated insulin levels can cause the downregulation of CBS and contribute to hyperhomocysteinemia [7,44]. We summarize this mechanism in Figure 6. In our study, we discovered that participants with high homocysteine levels tended to have metabolic syndrome, diabetes mellitus, and a greater waist circumference. Adipocytes, the cells responsible for insulin resistance and hyperinsulinemia, are often excessive in individuals with larger waist circumferences, diabetes mellitus, and metabolic syndrome. The results of our study support this mechanism, which may explain the association of metabolic syndrome with cardiovascular disease. Additionally, we confirmed that homocysteine levels were an independent risk factor for metabolic syndrome, even after considering all related factors in our study.

There are several strengths in our study. Aging is a significant risk factor for cardiometabolic diseases; thus, aged populations are vulnerable to cardiovascular disease. Our research focuses on middle-aged and older populations. Moreover, our participants were recruited from the community rather than patients with many comorbidities in the medical center. Our study can truly represent the relationship between homocysteine levels and cardiovascular diseases in the community’s middle-aged and older adult population. The results can be used as a reference for primary care. Other strengths of our study include sufficient sample size, straightforward study design, relative and sufficient confounders, appropriate analysis, and detailed discussion. Nevertheless, our study has some limitations. First, the homocysteine levels between groups were similar; therefore, future studies might need a larger sample size to explore clinical significance. Second, vitamin B deficiency is related to high homocysteine, but we did not evaluate vitamin B deficiency in our study. Third, we recruited our participants only from northern Taiwan. The selection bias in our study should be considered.

## 5. Conclusions

In this study, homocysteine was found to be an independent risk factor for MetS in middle-aged and elderly people in Taiwan. Thus, our results may provide valuable information for primary care physicians to alert subjects in this age group regarding an increased risk of MetS.

## Figures and Tables

**Figure 1 biomedicines-11-00378-f001:**
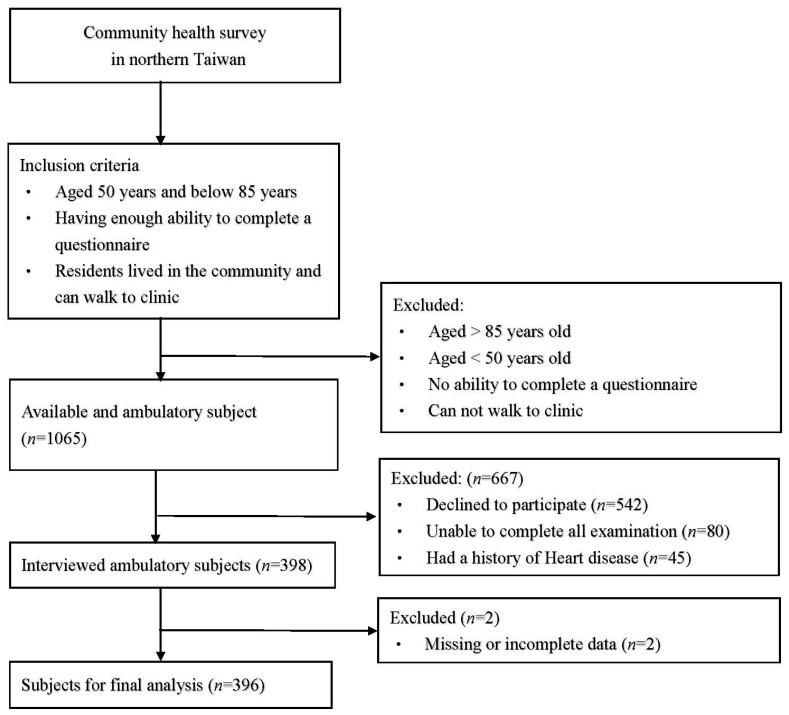
Flowchart of recruitment.

**Figure 2 biomedicines-11-00378-f002:**
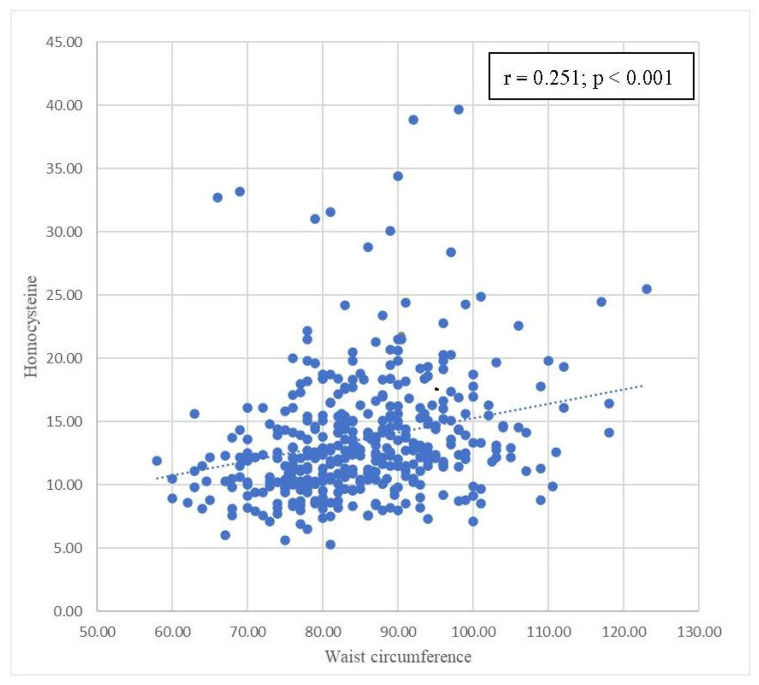
Scatterplot with the fit line of homocysteine levels by waist circumference.

**Figure 3 biomedicines-11-00378-f003:**
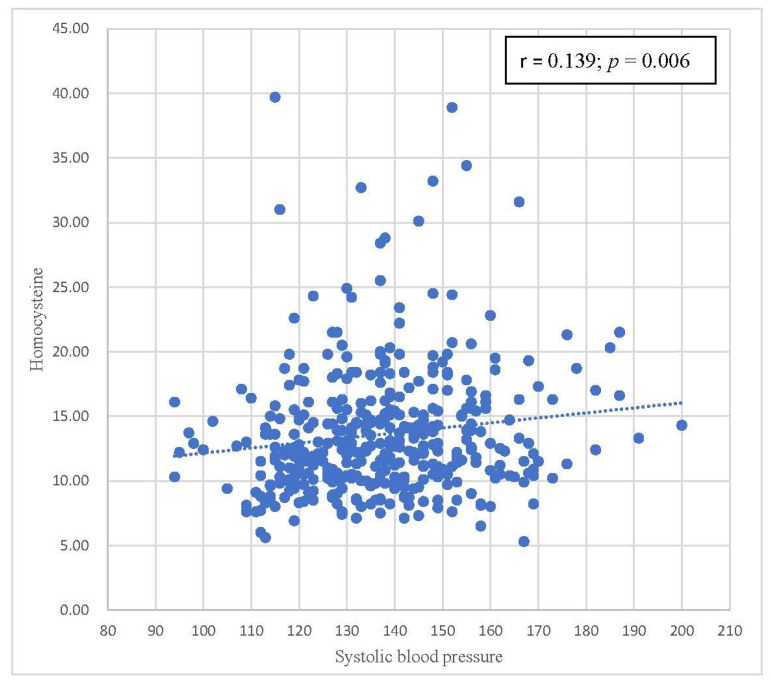
Scatterplot with the fit line of homocysteine levels by systolic blood pressure.

**Figure 4 biomedicines-11-00378-f004:**
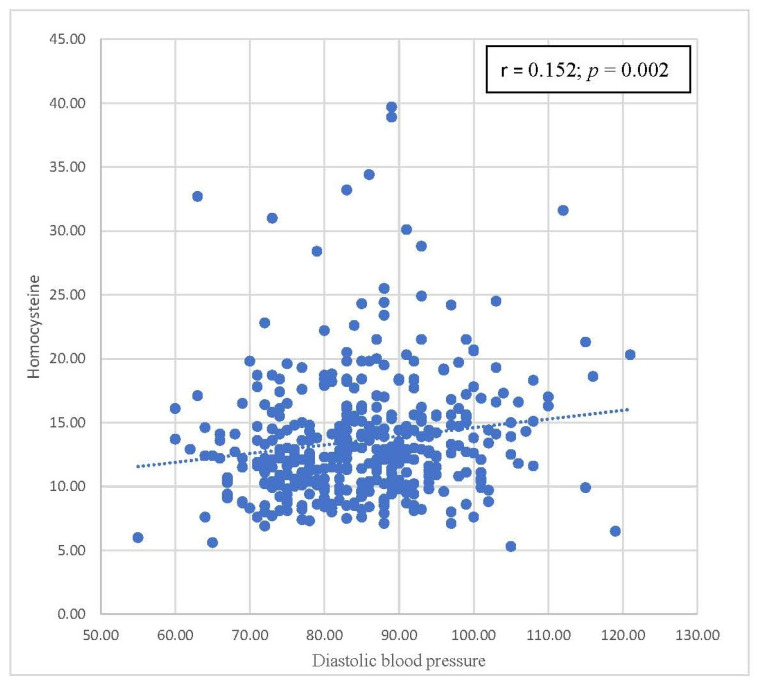
Scatterplot with the fit line of homocysteine levels by diastolic blood pressure.

**Figure 5 biomedicines-11-00378-f005:**
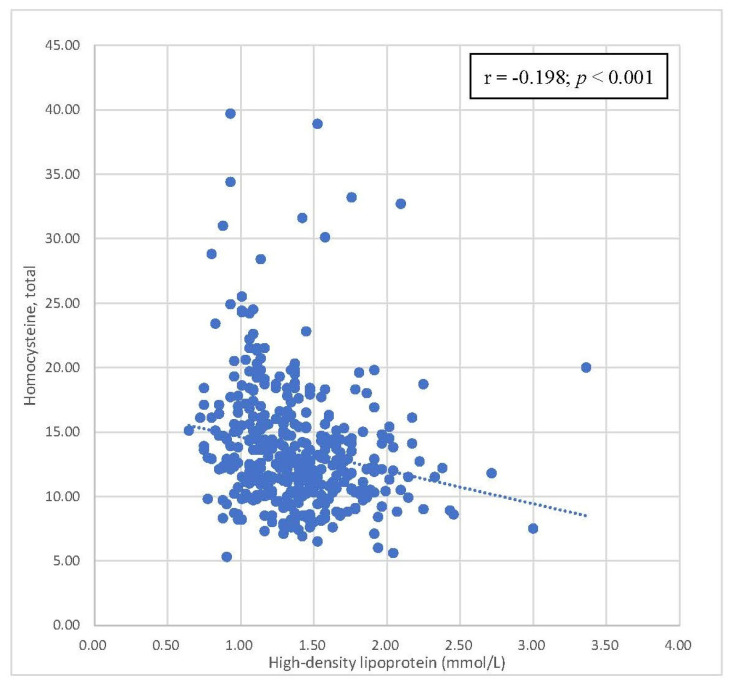
Scatterplot with the fit line of homocysteine levels by high-density lipoprotein.

**Figure 6 biomedicines-11-00378-f006:**
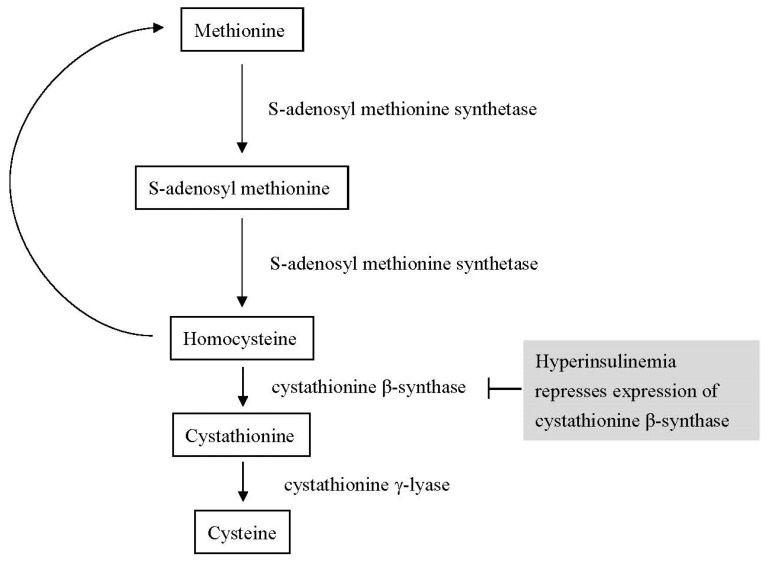
Interaction between homocysteine metabolism and hyperinsulinemia.

**Table 1 biomedicines-11-00378-t001:** The characteristics of study subjects by metabolic syndrome status.

Please Be Variable	Total	Nonmetabolic Syndrome Group	Metabolic Syndrome Group	*p*-Value
	(*n* = 396)	(*n* = 213)	(*n* = 183)	
Homocysteine (µmol/L)	13.60 ± 4.90	12.97 ± 4.64	14.34 ± 5.10	<0.001
FPG (mg/dL)	109.79 ± 35.65	99.27 ± 30.97	122.02 ± 36.90	<0.001
ALT (U/L)	27.16 ± 22.82	25.06 ± 20.85	29.60 ± 24.76	<0.001
Creatinine (mg/dL)	0.87 ± 0.43	0.85 ± 0.54	0.89 ± 0.26	<0.001
HDL-C(mg/dL)	53.65 ± 14.66	58.77 ± 14.09	47.49 ± 12.54	<0.001
LDL-C (mg/dL)	109.69 ± 33.99	113.36 ± 33.55	105.42 ± 34.10	<0.001
Triglycerides (mg/dL)	141.07 ± 110.00	101.75 ± 41.20	186.85 ± 142.72	0.006
Uric acid (mg/dL)	5.63 ± 1.52	5.32 ± 1.34	6.00 ± 1.63	0.006
Waist circumference (cm)	85.36 ± 10.83	81.12 ± 10.01	90.28 ± 9.62	0.048
SBP (mmHg)	137.30 ± 17.49	134.07 ± 17.62	141.07 ± 16.61	0.051
DBP (mmHg)	85.19 ± 10.98	83.37 ± 10.91	87.31 ± 10.70	0.371
Age (year)	64.80 ± 8.77	63.23 ± 9.38	64.28 ± 7.96	0.349
BMI (kg/m^2^)	25.59 ± 3.84	24.00 ± 3.10	27.45 ± 3.79	0.001
Smoking, *n* (%)	50 (12.6)	21 (9.9)	29 (15.8)	0.074
Alcohol consumption, *n* (%)	28 (7.1)	10 (4.7)	18 (9.8)	0.047
Hypertension, *n* (%)	201 (50.8)	71 (33.3)	130 (71.0)	<0.001
Diabetes mellitus, *n* (%)	133 (33.6)	40 (18.8)	93 (50.8)	<0.001
Dyslipidemia, *n* (%)	153 (38.6)	57 (26.8)	96 (52.5)	<0.001
Gender (men), *n* (%)	164 (41.4)	83 (39.0)	81 (44.3)	0.287

Note: Data are expressed as the mean ± SD for continuous variables and *n* (%) for categorical variables. Abbreviations: FPG, fasting plasma glucose; HDL-C, high-density lipoprotein; LDL-C, low-density lipoprotein; SBP, systolic blood pressure; DBP, diastolic blood pressure; BMI, body mass index.

**Table 2 biomedicines-11-00378-t002:** The percentage of participants who met the criteria for metabolic syndrome.

Variable	Nonmetabolic Syndrome (*n* = 213)	Metabolic Syndrome (*n* = 183)	*p*-Value
High waist circumference, *n* (%)	67 (31.5%)	146 (79.8%)	<0.001
High triglyceride, *n* (%)	19 (8.9%)	108 (59.0%)	<0.001
Low HDL-C, *n* (%)	10 (4.7%)	91 (49.7%)	<0.001
High BP, *n* (%)	149 (70.0%)	179 (97.8%)	<0.001
High FPG, *n* (%)	54 (25.4%)	142 (77.6%)	<0.001

Note: The following criteria are based on the definition of metabolic syndrome: (1) high WC: WC ≥ 90 cm for men and ≥80 cm for women; (2) high triglycerides: triglycerides ≥ 150 mg/dL; (3) low HDL-C: HDL-C < 40 mg/dL for men and <50 mg/dL for women; (4) high BP: BP ≥ 130/85 mmHg or current use of antihypertensive medications; and (5) high FPG: FPG ≥ 100 mg/dL. Abbreviations: FPG, fasting plasma glucose; HDL-C, high-density lipoprotein cholesterol; BP, blood pressure.

**Table 3 biomedicines-11-00378-t003:** The characteristics of study subjects according to tertiles of homocysteine.

	Homocysteine				
Variable	Total	First Group	Second Group	Third Group	*p*-Value
		(<11.1)	(11.1~14.3)	(≥14.4)	
	(*n* = 396)	(*n* = 132)	(*n* = 130)	(*n* = 134)	
Homocysteine (µmol/L)	13.60 ± 4.90	9.37 ± 1.26	12.66 ± 0.90	18.69 ± 4.92	<0.001
FPG (mg/dL)	109.79 ± 35.65	102.89 ± 20.65	111.74 ± 41.87	114.68 ± 39.74	0.007
ALT (U/L)	27.16 ± 22.82	26.06 ± 25.70	28.15 ± 21.44	27.28 ± 21.18	0.665
Creatinine (mg/dL)	0.87 ± 0.43	0.73 ± 0.14	0.80 ± 0.19	1.07 ± 0.66	<0.001
HDL-C(mg/dL)	53.65 ± 14.66	56.74 ± 13.89	54.42 ± 14.28	49.40 ± 14.28	<0.001
LDL-C (mg/dL)	109.69 ± 33.99	113.27 ± 32.46	108.09 ± 35.97	107.71 ± 33.46	0.183
TG (mg/dL)	141.07 ± 110.00	137.95 ± 105.33	141.65 ± 130.97	143.59 ± 91.41	0.677
Uric acid (mg/dL)	5.63 ± 1.52	5.13 ± 1.27	5.53 ± 1.35	6.23 ± 1.70	<0.001
Waist circumference (cm)	85.36 ± 10.83	81.01 ± 10.06	86.01 ± 10.55	89.00 ± 10.40	<0.001
SBP (mmHg)	137.30 ± 17.49	134.28 ± 16.60	137.29 ± 18.17	140.29 ± 17.29	0.005
DBP (mmHg)	85.19 ± 10.98	83.61 ± 10.62	84.04 ± 10.35	87.86 ± 11.49	0.001
Age (year)	64.80 ± 8.77	64.19 ± 8.50	65.34 ± 9.36	64.88 ± 8.47	0.521
BMI (kg/m^2^)	25.59 ± 3.84	24.53 ± 3.38	25.49 ± 3.54	26.74 ± 4.23	0.524
Smoking, *n* (%)	50 (12.6)	7 (5.3)	14 (10.8)	29 (21.6)	<0.001
Alcohol consumption, *n* (%)	28 (7.1)	8 (6.1)	9 (6.9)	11 (8.2)	0.495
HTN, *n* (%)	201 (50.8)	55 (41.7)	63 (48.5)	83 (61.9)	<0.001
DM, *n* (%)	133 (33.6)	29 (22.0)	49 (37.7)	55 (41.0)	<0.001
Dyslipidemia, *n* (%)	153 (38.6)	49 (37.1)	43 (33.1)	61 (45.5)	0.158
Metabolic syndrome, *n* (%)	183 (46.2)	48 (36.4)	60 (46.2)	75 (56.0)	<0.001
Gender (male), *n* (%)	164 (41.4)	24 (18.2)	52 (40.0)	88 (65.7)	<0.001

Note: The first group was defined as having homocysteine levels < 11.1 µmol/L, the second group was defined as having homocysteine levels between 11.1 and 14.3 µmol/L, and the third group was defined as having homocysteine levels ≥ 14.4 µmol/L. Data are expressed as the means ± SD for continuous variables and *n* (%) for categorical variables. Abbreviations: FPG, fasting plasma glucose; HDL-C, high-density lipoprotein cholesterol; LDL-C, low-density lipoprotein cholesterol; SBP, systolic blood pressure; DBP, diastolic blood pressure; BMI, body mass index.

**Table 4 biomedicines-11-00378-t004:** Pearson’s correlation between homocysteine and components of metabolic syndrome.

Variables	Homocysteine	
	Correlation Coefficient, r	*p*-Value
FPG (mg/dL)	0.091	0.069
Waist circumference (cm)	0.251	<0.001
SBP (mmHg)	0.139	0.006
DBP (mmHg)	0.152	0.002
Triglycerides (mg/dL)	0.024	0.627
HDL-C (mg/dL)	−0.198	<0.001

Abbreviations: FPG, fasting plasma glucose; SBP, systolic blood pressure; DBP, diastolic blood pressure; TG, triglyceride; HDL-C, high-density lipoprotein.

**Table 5 biomedicines-11-00378-t005:** Multiple logistic regression analysis of the association between homocysteine level and metabolic syndrome.

Variable	Model 1			Model 2			Model 3		
	OR	(95% C.I.)	*p* Value	OR	(95% C.I.)	*p* Value	OR	(95% C.I.)	*p* Value
First	1.00		0.006	1.00		0.014	1.00		0.08
Second	1.50	0.91 to 2.46	0.108	1.48	0.89 to 2.46	0.126	1.52	0.79 to 2.93	0.21
Third	2.22	1.36 to 3.64	0.001	2.23	1.30 to 3.81	0.004	2.32	1.12 to 4.78	0.02

Model 1 was unadjusted; Model 2 was adjusted for age and gender; and Model 3 was adjusted for age, sex, smoking, alcohol consumption, triglycerides, systolic blood pressure, and fasting plasma glucose.

## Data Availability

The original contributions presented in the study are included in the article. Further requirements can be directed to the corresponding author.

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
