# Peer review of "The Relationship between Elevated Homocysteine and Metabolic Syndrome in a Community-Dwelling Middle-Aged and Elderly Population in Taiwan"

_biomedicines, 2023, doi:10.3390/biomedicines11020378_

Round 1

Reviewer 1 Report

The study by Shih et al. looked into the plasma level of homocysteine as a potential biomarker for cardiovascular diseases. The study was focused on a middle-aged and elderly population in Taiwan. Plasma homocysteine concentrations, along with several other metabolic indices, were compared between the population with metabolic syndrome (MetS) and that without the metabolic abnormalities. The authors concluded that high levels of plasma homocysteine correlate well with the presence of metabolic syndrome and increased risks of cardiovascular diseases. The authors also suggested that primary care physicians may use the elevated levels of plasma homocysteine as an index for increased risks of metabolic syndrome in patients.

While abnormal plasma concentrations of metabolite biomarkers, like those found in hyperhomocysteinemia, could be beneficial for the diagnosis of diseases, this study has not well demonstrated that plasma homocysteine concentrations could become one of the biomarkers. The conclusion is not well supported by the results. Data analyses are not rigorous. There are also several flaws in this paper that need to be addressed.

It appears that routinely measured metabolic indices such as plasma glucose, TG, LDL etc. offer clear differences between the healthy population and that with metabolic syndrome. This is of course expected as these indices have been gold-standards for assessing metabolic states in patients. Also, this is not to say a new metabolic index like plasma homocysteine proposed in this article should not be considered. The plasma homocysteine levels between the two groups reported here is quite similar. A much larger population sample is needed in order to ascertain whether 12.9 vs 14.3 micromoles per liter here is really different.

According to several guidelines, plasma homocysteine levels of over 50 micromoles per liter is considered high, posing cardiovascular risks. In this study, the level of 14 micromoles per liter (~3.5-fold lower) is considered in the high group. What guidelines did the authors use to classify this value as a high level? Also, several articles classify a level of lower than 15 micromoles per liter as normal.

Also, it is not clear what guidelines were used to classify plasma homocysteine levels shown in Table as low (<11 micromol/L), middle (11-14 micromol/L), and high (>14 micromol/L). Again, compared to the two guidelines mentioned above, all these values should be considered normal.

It is known that high levels of plasma homocysteine are associated with vitamin B deficiencies. Were vitamin deficiencies considered in this study?

Minor comments:

Figure 1 has different fonts than the rest of the text.

There are several grammatical errors throughout the manuscript that need to be addressed. For examples, criteria includes (p.1), which produced by (p.1), are wall developed (p.1), was showed (p.5), three model was applied (p.5) etc.

The unit “umol/L” in Tables 1 and 2 should be written with a Greek letter instead of a lower case u.

The unit liter in deciliter should be written as dL instead of dl (Tables 1 & 2).

The use of abbreviations is excessive, especially in the main text. Given the relatively large number of abbreviations used here, it is quite difficult to follow.

Author Response

Dear Reviewer

Due to the new year's holiday, our English editing faced severe delay. Thank the reviewer for reviewing and consideration. Here we respond to the comment point-to-point:

The study by Shih et al. looked into the plasma level of homocysteine as a potential biomarker for cardiovascular diseases. The study was focused on a middle-aged and elderly population in Taiwan. Plasma homocysteine concentrations, along with several other metabolic indices, were compared between the population with metabolic syndrome (MetS) and that without the metabolic abnormalities. The authors concluded that high levels of plasma homocysteine correlate well with the presence of metabolic syndrome and increased risks of cardiovascular diseases. The authors also suggested that primary care physicians may use the elevated levels of plasma homocysteine as an index for increased risks of metabolic syndrome in patients.

COMMENT 1

While abnormal plasma concentrations of metabolite biomarkers, like those found in hyperhomocysteinemia, could be beneficial for the diagnosis of diseases, this study has not well demonstrated that plasma homocysteine concentrations could become one of the biomarkers. The conclusion is not well supported by the results. Data analyses are not rigorous. There are also several flaws in this paper that need to be addressed.

RESPONSE 1

We thank the reviewer for the comment. Our paper underwent a major revision with English editing. We present the latest version in our reply.

COMMENT 2

It appears that routinely measured metabolic indices such as plasma glucose, TG, LDL etc. offer clear differences between the healthy population and that with metabolic syndrome. This is of course expected as these indices have been gold-standards for assessing metabolic states in patients. Also, this is not to say a new metabolic index like plasma homocysteine proposed in this article should not be considered. The plasma homocysteine levels between the two groups reported here is quite similar. A much larger population sample is needed in order to ascertain whether 12.9 vs 14.3 micromoles per liter here is really different.

RESPONSE 2

We value the concern of the reviewer. Although, our research revealed the relationship between homocysteine level and metabolic syndrome. Further research with a bigger sample size should be performed to explore clinical significance. We added the reviewer’s concern to our limitation as below:

“Nevertheless, our study has some limitations. First, the homocysteine levels between groups were similar; therefore, future studies might need a larger sample size to explore clinical significance.” (line: 350~352)

COMMENT 3

According to several guidelines, plasma homocysteine levels of over 50 micromoles per liter is considered high, posing cardiovascular risks. In this study, the level of 14 micromoles per liter (~3.5-fold lower) is considered in the high group. What guidelines did the authors use to classify this value as a high level? Also, several articles classify a level of lower than 15 micromoles per liter as normal. Also, it is not clear what guidelines were used to classify plasma homocysteine levels shown in Table as low (<11 micromol/L), middle (11-14 micromol/L), and high (>14 micromol/L). Again, compared to the two guidelines mentioned above, all these values should be considered normal.

RESPONSE 3

We thank the reviewer for allowing us to further explain this issue. We classified our participants into three groups according to the tertile of homocysteine levels from low to high rather than any existed criteria. In order to avoid confusion, we replaced low, middle, and high homocysteine groups with first, second, and third groups respectively throughout our article. We also added the description in Materials and Methods as below:

“The participants were categorized into the first group (homocysteine <11.1 µmol/L), second group (homocysteine between 11.1 and 14.3 µmol/L), and third group (homocysteine ≥14.4 µmol/L) according to the tertile of homocysteine levels from low too high in our study.” (line: 89~92)

COMMENT 4

It is known that high levels of plasma homocysteine are associated with vitamin B deficiencies. Were vitamin deficiencies considered in this study?

RESPONSE 4

We fully agree with the reviewer’s concern. However, we do not evaluate the vitamin B deficiencies in our research. We recognize this issue as our flaw, and we listed it in our limitation as below:

“Second, vitamin B deficiency is related to high homocysteine, but we did not evaluate vitamin B deficiency in our study..” (line: 353~354)

Minor comments:

COMMENT 5

Figure 1 has different fonts than the rest of the text.

RESPONSE 5

We thank reviewer for the reminding, and we corrected the error

COMMENT 6

There are several grammatical errors throughout the manuscript that need to be addressed. For examples, criteria includes (p.1), which produced by (p.1), are wall developed (p.1), was showed (p.5), three model was applied (p.5) etc.

RESPONSE 6

We thank the reviewer for the notification. The article underwent English editing by American Journal Experts (AJE) after we revised the article

COMMENT 7

The unit “umol/L” in Tables 1 and 2 should be written with a Greek letter instead of a lower case u.

RESPONSE 7

We thank reviewer for the reminding, and we corrected the error

COMMENT 8

The unit liter in deciliter should be written as dL instead of dl (Tables 1 & 2).

RESPONSE 8

We thank reviewer for the reminding, and we corrected the error

COMMENT 9

The use of abbreviations is excessive, especially in the main text. Given the relatively large number of abbreviations used here, it is quite difficult to follow.

RESPONSE 9

We thank the reviewer for the suggestion. In order to reduce the abbreviation in our article, we canceled the abbreviation of some terms, including metabolic syndrome, triglyceride, hypertension, and diabetes mellitus, throughout our article.

Reviewer 2 Report

Shih et al. show in an aging  Taiwanese cohort that homocystene levels closely associate with metabolic syndrome. The paper has clear study design, a sizeable cohort and provides additional links between homocysteine and metabolic syndrome in a cohort where this has never been shown.

In general there are several language spelling and mistake errors that make it hard to read. An extensive editing in the English language is needed.

Major comments:

Figure 1 is not well described and redundant since it data is already shown in table 2. If the authors want to show individual level data, they could show scatter plots of the highly significant correlations in Table 3 (WC vs homocysteine, HDL vs homocysteine).

Minor comments:

Introduction

* The gap in knowledge of why they performed the study is missing.

* Missing link to homocysteine and its mechanisms of action (to be moved from the discussion)

* In general, the discussion include introductory info that should be included in the introduction.

Methods:

how were the participants recruited? 

Results:

Add a separate table on which criteria for metabolic syndrome was met for each participant should be included in the MetS group (% of MetS individuals who had high TG, % of MetS individuals with high BP, etc). In fact, it is odd that there is no difference in SBP between metabolic syndrome individuals and controls.

Table 3: Why is there a p-value of the reference group for HomoCysteine? By definition the control group should not have a p-value (only incomparison with the other two groups)

Author Response

Dear Reviewer

Due to the new year's holiday, our English editing faced severe delay. Thank the reviewer for reviewing and consideration. Here we respond to the comment point-to-point.

Shih et al. show in an aging  Taiwanese cohort that homocystene levels closely associate with metabolic syndrome. The paper has clear study design, a sizeable cohort and provides additional links between homocysteine and metabolic syndrome in a cohort where this has never been shown.

COMMENT 1

In general there are several language spelling and mistake errors that make it hard to read. An extensive editing in the English language is needed.

RESPONSE 1

We thank the reviewer for the notification. The article underwent English editing by American Journal Experts (AJE) after we revised the article

Major comments:

COMMENT 2

Figure 1 is not well described and redundant since it data is already shown in table 2. If the authors want to show individual level data, they could show scatter plots of the highly significant correlations in Table 3 (WC vs homocysteine, HDL vs homocysteine).

RESPONSE 2

The suggestion from the reviewer was valuable, and this was fully taken. We presented scatter plots of all significant correlations in Table 3 as figure 2, figure 3, figure 4, and figure 5.

Minor comments:

Introduction

COMMENT 3

* The gap in knowledge of why they performed the study is missing.

RESPONSE 3

We thank the reviewer for allowing us to further explain this issue. There are several advantages of our research, and we add it to our discussion as below:

“There are several strengths in our study. Aging is a significant risk factor for cardiometabolic diseases; thus, aged populations are vulnerable to cardiovascular disease. Our research focuses on middle-aged and older populations. Moreover, our participants were recruited from the community rather than patients with many comorbidities in the medical center. Our study can truly represent the relationship between homocysteine levels and cardiovascular diseases in the community's middle-aged and older adult population. The results can be used as a reference for primary care. Other strengths of our study include a sufficient sample size, straightforward study design, relative and sufficient confounders, appropriate analysis, and detailed discussion.” (line: 342~350)

COMMENT 4

* Missing link to homocysteine and its mechanisms of action (to be moved from the discussion)

RESPONSE 4

We thank the reviewer for the valuable suggestion. The related paragraph was in discussion, and we also expanded the content as below:

“In our study, we discovered that participants with high homocysteine levels tended to have metabolic syndrome, diabetes mellitus, and a greater waist circumference. Adipocytes, the cells responsible for insulin resistance and hyperinsulinemia, are often excessive in individuals with a larger waist circumferences, diabetes mellitus, and metabolic syndrome. The results of our study support this mechanism, which may explain the association of metabolic syndrome with cardiovascular disease.” (line: 333~339)

COMMENT 5

* In general, the discussion include introductory info that should be included in the introduction.

RESPONSE 5

The suggestion from the reviewer is constructive. We modified the content and added it in the introduction as below:

“A recent study revealed that elevated insulin can disturb the metabolism of homocysteine and cause elevated homocysteine levels [7]. Insulin resistance in metabolic syndrome is marked by high insulin levels [8]. This evidence indicates that homocysteine may play an important role in the relationship between metabolic syndrome and cardiovascular disease. Additionally, elderly populations are vulnerable to cardiometabolic diseases [9]. Hence, in our study, we investigated the relationship between homocysteine levels and metabolic syndrome among the middle-aged and elderly population in Taiwan.” (line: 42~49)

Methods:

COMMENT 6

how were the participants recruited? 

RESPONSE 6

The concern from the reviewer is crucial. In response to the reviewer, we added the flowchart and description in Materials and Methods. Figure 1 shows the flowchart of recruitment.

Figure 1 (in manuscript)

Results:

COMMENT 7

Add a separate table on which criteria for metabolic syndrome was met for each participant should be included in the MetS group (% of MetS individuals who had high TG, % of MetS individuals with high BP, etc). In fact, it is odd that there is no difference in SBP between metabolic syndrome individuals and controls.

RESPONSE 7

We thank the reviewer for the suggestion. We compared HDL, WC, BP, FPG, and triglycerides between MetS and non-MetS group. The result was shown in Table 2B and related-content was also added as below:

“Based on the definition of metabolic syndrome [7], we compared the criteria of metabolic syndrome in both groups using the chi-square test. The results are shown in Table 1B. As we expected, the metabolic syndrome group had significantly higher WC, BP, FPG, and triglycerides than the nonmetabolic group. In addition, the metabolic group had significantly lower HDL-C levels than the nonmetabolic syndrome group.” (line: 132~137)

Variable

Nonmetabolic syndrome

(n=213)

Nonmetabolic syndrome

(n=213)

p value

High waist circumference, n (%)

67 (31.5%)

146 (79.8%)

<0.001

High triglyceride, n (%)

19 (8.9%)

108 (59.0%)

<0.001

Low HDL-C, n (%)

10 (4.7%)

91 (49.7%)

<0.001

High BP, n (%)

149 (70.0%)

179 (97.8%)

<0.001

High FPG, n (%)

54 (25.4%)

142 (77.6%)

<0.001

Table 1B. The percentage of participants who met the criteria for metabolic syndrome

Note: The following criteria are based on the definition of metabolic syndrome (1) High WC: WC ≥ 90 cm for men and ≥ 80 cm for women; (2) High triglycerides: triglycerides ≥ 150 mg/dL; (3) Low HDL-C: HDL-C < 40 mg/dL for men and < 50 mg/dL for women; (4) High BP: BP ≥ 130/85 mmHg or current use of antihypertensive medications; and (5) High FPG: FPG ≥ 100 mg/dL.

Abbreviations: FPG, fasting plasma glucose; HDL-C, high-density lipoprotein cholesterol; BP, blood pressure

COMMENT 8

Table 3: Why is there a p-value of the reference group for HomoCysteine? By definition the control group should not have a p-value (only incomparison with the other two groups)

RESPONSE 8

Thank the reviewer for the question. We used Pearson’s correlation in Table 3, so every variable has p-value

Reviewer 3 Report

The manuscript entitled Relationship between hyperhomocysteinemia and metabolic 2 syndrome in community-dwelling middle-aged and elderly 3 population in Taiwan, is an original paper. This is well written. It has, also, practical implications. I have some remarks/comments and unclear thinks.

Alcohol consumption was significantly higher in metabolic syndrome group (table 10). Do you think that it could interfere with the results of this study?

Best Pearson’s correlation between homocysteine and components of metabolic syndrome was with waist circumference (which means abdominal obesity). This is an important message in this study. Why do you think?

Page 7 line 37-50: I suggest making a schema/figure with homocysteine biosynthesis and possible relationship with insulin resistance and MetS.

Author Response

Dear Reviewer

Due to the new year's holiday, our English editing faced severe delay. Thank the reviewer for reviewing and consideration. Here we respond to the comment point-to-point.

COMMENT 1

The manuscript entitled Relationship between hyperhomocysteinemia and metabolic 2 syndrome in community-dwelling middle-aged and elderly 3 population in Taiwan, is an original paper. This is well written. It has, also, practical implications. I have some remarks/comments and unclear thinks.

RESPONSE 1

We appreciate the reviewer’s comment and thank the reviewer for reviewing our article.

COMMENT 2

Alcohol consumption was significantly higher in metabolic syndrome group (table 10). Do you think that it could interfere with the results of this study?

RESPONSE 2

We thank the reviewer for the comment. We did not have table 10 in our article, but we noted that alcohol consumption has a significant difference in Table 1. Hence, we included alcohol consumption in the model 3 in Table 4 to evaluate the influence of alcohol consumption. The odds ratio for metabolic syndrome was 2.32 (1.12 to 4.78) with a p-value of 0.02 in the high homocysteine group comparing to the low homocysteine group.

COMMENT 3

Best Pearson’s correlation between homocysteine and components of metabolic syndrome was with waist circumference (which means abdominal obesity). This is an important message in this study. Why do you think?

RESPONSE 3

We thank the reviewer for allowing us to explain this issue further. High waist circumference indicates central obesity, which also has a close relationship with insulin resistance. The high insulin level in insulin resistance can also suppress the CBS and lead to high homocyateine. We added this concept after we discussed the mechanism of CBS in the discussion as below:

“In our study, we discovered that participants with high homocysteine levels tended to have metabolic syndrome, diabetes mellitus, and a greater waist circumference. Adipocytes, the cells responsible for insulin resistance and hyperinsulinemia, are often excessive in individuals with a larger waist circumferences, diabetes mellitus, and metabolic syndrome. The results of our study support this mechanism, which may explain the association of metabolic syndrome with cardiovascular disease.” (line: 333~339)

COMMENT 4

Page 7 line 37-50: I suggest making a schema/figure with homocysteine biosynthesis and possible relationship with insulin resistance and MetS.

RESPONSE 4

We thank the reviewer for the suggestion. In order to enunciate the relationship, we present Figure 5 to describe the relationship[ between hyperinsulinemia and homocysteine metabolism.

Figure 5. Relationship between homocysteine metabolism and insulin (in manuscript)

Round 2

Reviewer 1 Report

The authors have responded to my concerns. Significant changes have also been made to the manuscript.

Reviewer 3 Report

Thank you for responding to my comments/suggestions. I am sorry for the editing error (table 1 not table 10). Figure 5 in your responses is acctually figure 6.